# Interferons—Implications in the Immune Response to Respiratory Viruses

**DOI:** 10.3390/microorganisms11092179

**Published:** 2023-08-29

**Authors:** Harrison C. Bergeron, Matthew R. Hansen, Ralph A. Tripp

**Affiliations:** Department of Infectious Diseases, University of Georgia College of Veterinary Medicine, Athens, GA 30605, USA; harrison.bergeron@uga.edu (H.C.B.); matthew.hansen@uga.edu (M.R.H.)

**Keywords:** respiratory syncytial virus, RSV, influenza, SARS-CoV-2, COVID-19, interferons, IFN, immunity

## Abstract

Interferons (IFN) are an assemblage of signaling proteins made and released by various host cells in response to stimuli, including viruses. Respiratory syncytial virus (RSV), influenza virus, and SARS-CoV-2 are major causes of respiratory disease that induce or antagonize IFN responses depending on various factors. In this review, the role and function of type I, II, and III IFN responses to respiratory virus infections are considered. In addition, the role of the viral proteins in modifying anti-viral immunity is noted, as are the specific IFN responses that underly the correlates of immunity and protection from disease.

## 1. Interferons 

Interferons (IFNs) were first discovered during a study on viral interference in egg chorioallantoic membranes. The early study found that a factor was released which had the ability to induce viral interference, hence “interferon” [1]. This soluble IFN was detected within 3 h of infection and was found to confer resistance to viral replication in treated cells before infection [1,2]. There are now three known types of IFNs: type I, II, and III, each with distinct and overlapping functions which signal through different receptors. Type I IFNs signal through the IFN alpha receptor (IFNAR), type II signaling occurs through the IFN gamma receptor (IFNGR), and type III IFN signaling occurs through the IFN lambda receptor (IFNLR) [1,3,4,5]. Type I IFN consists of 14 subtypes of IFNα, INFß, and IFNκ and IFNω [6]. Most cell types are capable of secreting IFNß, but plasmacytoid dendritic cells (pDCs) and other innate immune cells are the primary producers of IFNα [7]. IFNγ is the only type II IFN [8]. Adaptive and innate immune cells, including T helper 1 (Th1) cells, CD8+ cytotoxic T lymphocytes (CTLs), natural killer (NK) cells, and innate lymphoid cells (ILCs), are responsible for the majority of IFNγ production [9]. IFNγ expression is linked to cell-mediated immunity against intracellular pathogens/viruses, macrophage activation and polarization towards an M1 phenotype, and IgG class switching [9,10]. Type III IFNs include four subtypes: IFN-λ1 (IL-29), IFN-λ2 and IFN-λ3 (IL-28A and IL28B, respectively), and IFN-λ4 [11]. IFNλ4 is the most recently discovered IFN. Type III IFNs are the first defense against virus replication in epithelial cells and are less likely to cause damaging inflammatory responses compared to the more potent type I IFN response. The IFNLR receptor is found in mucosal sites, inducing less potent inflammatory properties [3,4,5]. The IFN response is crucial in limiting viral replication and dissemination [12].

## 2. Downstream Signaling

The development of innate and adaptive antiviral immunity is influenced by pattern recognition receptors (PRRs). The activation of innate immunity by respiratory viruses can occur either on the cell surface through host cell receptors or by the interaction of intracellular and cytosolic PRRs. These receptors detect viral components such as RNA or DNA, resulting in the generation of downstream pathways that lead to antiviral immunity [13]. When PRRs recognize pathogen-associated molecular patterns (PAMPs), type I and III IFN responses are initiated which activate IFN regulatory factors (IRFs) in a signaling cascade [14]. Toll-like receptors (TLRs), retinoic acid-inducible gene I (*RIG-1*) receptors, and nucleotide-binding oligomerization domain (NOD)-like receptors (NLRs) are the principal IFN-stimulating PRRs [14,15,16]. TLR recognition of viral PAMPs signals through myeloid differentiation primary response 88 (MYD88) which acts as an adapter connecting proteins that receive signals from outside the cell to the proteins that relay signals inside the cell [17]. RIG-like receptors (RLRs) primarily detect viral genomes [18] and signal through the mitochondrial antiviral signaling protein (MAVS) and TIR domain-containing adaptor inducing INFß (TRIF) [19]. The binding of viral PAMPs ultimately leads to the activation of IRF3 and IRF7, which drive the transcription of type I and type III IFNs [20]. These pathways also activate nuclear factor κ-B (NF-κB) leading to the transcription of other inflammatory cytokines [14,21]. Binding of type I and III IFNs results in the recruitment of signal transducer and activator of transcription (STAT) for phosphorylation by Janus kinases (JAKs) [22]. The JAK/STAT pathway is the major signaling pathway activated by IFNs, leading to the expression of IFN-stimulated genes (ISGs). To reduce IFN effector function, viruses have developed various strategies to antagonize the JAK/STAT pathway [23].

The primary function of ISGs is to detect viral RNA, thus inhibiting viral replication [24]. The activation of the genes responsible for producing IFNs creates a positive feedback loop. ISGs, including transcription factors, have the ability to activate multiple genes. Differential activation is associated with the various subtypes of type I and type III IFNs and is believed to occur due to varying binding affinities of the IFN subtypes and the shared receptor. The outcomes of these activations are still being studied [5,25]. In response to viral DNA and RNA, the *IFIT* family of ISGs is commonly induced by type I and type III IFNs. *IFIT* genes suppress viral infection primarily by limiting viral RNA and DNA replication and impairing the entry of enveloped viruses [26]. Oligoadenylate synthetase (OAS1) functions in the activation of RNase L which cleaves viral RNA species [27]. ISG15 is a ubiquitin-like protein that can target viral proteases, preventing replication [28]. ISG20 is a 3′-5′ exonuclease which can cleave viral RNA, thereby reducing replication and triggering PRRs [29]. Viperin functions by disrupting lipid rafts which interferes with viral replication complexes [30]. As multiple ISGs exist, we direct readers to a recently published comprehensive review of ISGs [24].

IFNγ is different from types I and III and has a unique structure and function. It is primarily secreted by activated CD4 T helper and CD8 T cells, natural killer (NK) cells, NK T cells, and professional antigen-presenting cells (APCs) [31]. The production of IFNγ is prompted by IL-12 and IL-18, which are primarily produced by macrophages and dendritic cells [32,33]. While Th1 CD4 cells are major producers of IFNγ, CD8 CTLs also express it [9]. IL-12 drives the Th1 differentiation of naïve CD4 T cells, which is necessary for the expression of IFNγ [34]. The expression of IFNγ relies on transcription factors such as STAT4, Tbet, AP-1, and Eomes [35,36,37]. When IFNγ binds to IFNGR it results in JAK phosphorylation of STAT1 which causes the formation of homodimers that act as transcription factors for γ-activated sites (GAS) [9]. GAS includes a multitude of genes, including transcription factors that drive the expression of other genes, similar to ISGs stimulated by type I and type III IFNs. When macrophages are activated by IFNγ it results in upregulated antigen presentation, increased sensitivity to cytokines and chemokines, increased sensitivity of PRRs, and decreased sensitivity to anti-inflammatory cytokines such as IL-10 [9]. IFNγ is essential in linking T cells and macrophages and cellular immunity to intracellular pathogens so any shortcomings in this response can lead to increased susceptibility [38,39].

## 3. IFNs in Early Life and Childhood

Infants initially have a Th17/Th2-biased immune response with lower levels of IFNs [40,41,42]. A study of a longitudinal cohort found that levels of type I (and type III) IFNs were reduced in cord blood but increased with age [43]. Interestingly, infants with low IFN levels at birth had a higher risk of developing severe respiratory tract infections and persistent wheezing later in life [43]. When stimulated, cord blood responses showed decreased IFNα expression compared to whole blood from adult donors, while IFNß production in neonatal samples was similar to monocytes and slightly increased to whole blood from adult donors [44]. Another study found that IFNß expression was impaired in cord blood and monocyte-derived dendritic cells in response to TLR4 stimulation, due to defective CREB binding protein (CBP) binding to IRF3 [45]. Similarly, pDC-dependent IFNα response in cord blood plasmacytoid dendritic cells (PDC) has been shown to be deficient in response to TLR9 stimulation due to defective nuclear translocation of IRF7 [46]. Infants, particularly pre-term ones, produce lower levels of IFNs after TLR9 stimulation compared to adults [47]. Researchers continue to investigate the role of type III IFNs in human development. A recent study conducted on primary human airway epithelium cultures (AECs) showed that infant AECs produce more IFNλ1 than young children when exposed to poly I:C stimulation [48]. The study also found that infants hospitalized with respiratory virus infections produce higher levels of IFNλ1 in nasal aspirate than young children [48]. Interestingly, the study also observed that IFNLR1/IL10RB upregulation was specific to certain pathogens in infants aged 1–6 months, and this was associated with more severe bronchiolitis and eosinophilia, implying that IFNλ signaling is connected to disease in certain cases [49].

At birth, the expression of IFNγ and other Th1-associated responses is reduced but increases as age progresses [50,51,52]. This pattern is observed in a mouse model of respiratory syncytial virus (RSV) infection as well. This could be attributed to the fact that the fetal liver gives rise to neonatal T cells, whereas in adults, it is the bone marrow [53]. In a mouse model that involved thymic transfer of fetal derived T cells into adult mice it was observed that fetal-derived T cells produced higher levels of cytokines than adult-derived T cells and were skewed towards a Th2-bias [53,54,55]. This effect maybe partly due to the lack of IL-12 production by dendritic cells (DCs) in neonatal mice, as IL-12 is a cytokine associated with differentiation of Th1 cells [56]. In addition, cord blood and moDCs stimulated with LPS showed impaired CBP and IRF binding compared to adults. This is relevant because IRF3 is a necessary transcription factor for IL-12 expression [45]. The epigenetic profile of neonatal T cells is linked to enhanced IL-4 and IL-13 expression, and the IL-13Rα1, in conjunction with IL4Rα acts as a receptor for IL-4 in a neonatal mouse model and induces apoptosis of Th1 cells further driving the Th2-bias [57,58,59]. GATA3, a Th2-associated transcription factor, was found to be upregulated in neonatal CD4+ T cells [58]. In human cord blood and neonatal adenoid tissue, IL4 receptor expression at an early stage in development decreased with age and may contribute to Th2-biasing [60]. All these studies support the conclusion that age-dependent immune responses to viral infections occur, with infants producing lower IFN levels and subsequent Th2 biasing.

## 4. Mice as a Model

Mice are often used in translational research due to their easy housing and relatively low cost, as well as the availability of a range of immunology reagents. However, IFN responses vary between strains of mice and between mice and humans. In mice, there is a notable difference in the expression of type III IFN compared to humans. Humans possess up to four functional subtypes of IFNλ, while mice only have functional IFNλ2/3 and a pseudogene for IFNλ1 [61,62]. Studies have shown that age-dependent differences in IFN response also exist in mice, which reflect those observed in humans. For instance, in the context of RSV, neonatal mice exhibit reduced IFNα expression by pDCs and a tendency towards Th2-type cytokines [63,64]. Furthermore, it has been shown that aged mice are more susceptible to severe SARS-CoV-2 disease, a feature linked to an impaired IFN and antibody response [65]. Although mouse models do recapitulate age-related discrepancies in the severity of respiratory virus infection observed in humans, it is essential to consider the biological differences between the two species to interpret the data accurately. Genetic susceptibility to infection is determined by defects in genes that control non-redundant pathways of type I and III IFN responses [66,67].

Certain inbred strains (e.g., C57/BL6 and BALB/c) lack a functional *Mx1* gene, which is a dominant antiviral resistance gene known as Mx for ‘myxovirus resistance’ [68,69]. Mice with the *Mx1* gene are able to survive infection with mouse-adapted influenza A virus at doses that would be lethal for standard inbred strains [70]. Interestingly, neonatal mice with a functioning *Mx1* gene remain just as susceptible to influenza challenge as their *Mx1*-negative counterparts [71]. However, when treated with exogenous IFN, *Mx1*-competent mice become resistant, similarly to their adult counterparts, whereas neonates without a functioning *Mx* gene do not respond to IFN treatment [72,73]. These experiments demonstrate the significance of IFN in the susceptibility of newborns and young children to influenza virus as modeled in neonatal mice [72,73]. Other mammals used in respiratory virus infection studies, such as cotton rats, hamsters, and ferrets, have functional *Mx* genes [74,75,76]. In humans, the *MxA* gene has an important role in inhibiting viral replication by interfering with the assembly of viral ribonucleoprotein, although the exact mechanism is yet to be determined [77,78].

## 5. Respiratory Syncytial Virus 

RSV is a virus that can cause serious respiratory tract diseases and death in certain groups of people, including young children (up to 60 months old), the elderly, and those with weakened immune systems [79]. A 2019 review of RSV cases around the world estimated that there were 3.6 million hospital admissions, 26,300 in-hospital deaths, and 101,400 total deaths attributable to RSV [80]. Appallingly, 45,700 of those deaths were infants between the ages of 0–6 months, accounting for 2% of all deaths in that age range [80]. RSV is a single strand of RNA and belongs to the Pneumoviridae family. Its genome is 15.2 kb long and contains 10 genes that produce 11 proteins. One of the genes, M2, actually produces two different proteins, M2.1 and M2.2 [81,82]. RSV’s genome contains two nonstructural (NS) proteins, NS1 and NS2, as well as nucleocapsid proteins N, L, and P, regulatory protein M2, the inner envelope protein M1, and three surface proteins SH, G, and F. The G and F proteins are the main antigenic proteins. RSV has two main lineages, A and B, which are defined by differences in their G protein sequences [81,82]. RSV usually infects ciliated airway epithelial cells [83]. RSV can cause a variety of health problems, including changes in lung structure, decreased lung function, and increased mucosal responses [79]. RSV’s proteins are known to cause a range of immune responses that can suppress the body’s antiviral response and even cause damage to the host’s immune system [82,84,85]. Several studies have shown that RSV can fail to elicit a strong type I IFN response [86,87,88,89].

Research has shown that the RSV NS1 and NS2 proteins are effective in suppressing the IFN response. Removing these proteins results in a weakened response making NS1/NS2 deletion mutants’ potential candidates for a vaccine [86,90,91]. NS1 and NS2 are known to inhibit type I IFNs by blocking STAT2 in human epithelial cells [92]. Although NS2 was originally believed to be redundant to NS1, recent studies have revealed significant differences between them, such as the lack of structural homology in the crystallographic structure of NS2 [93]. NS2 has been shown to inhibit *RIG-1* activation by binding to the N-terminal caspase activation and recruitment domain (CARD) of *RIG-1*, thus preventing downstream interaction with MAVS [94]. In contrast, NS1 binds to MAVS and inhibits its interaction with *RIG-1* [93,95]. In mice, alveolar macrophages induced by MAVS coupled with RLRs are the primary source of type I IFNs during RSV infection [96]. Overall, the NS proteins block separate signaling proteins within identical pathways, resulting in a weak IFN response to infection.

The RSV G protein has been found to affect the immune response through a CX3C motif that is present in all RSV strains and is similar to the natural chemokine fractalkine [97]. The G protein inhibits the production of type I IFN by interacting with the CX3CR1 receptor on pDCs and monocytes through its CX3C motif. Blocking this interaction through a CX4C mutant virus or mAb treatment that targets the interaction increased IFNα and pro-inflammatory cytokines such as TNFα [85,98]. A recent study that examined anti-RSV G monoclonal antibodies (mAbs) 3D3 and 2D10 found that treatment improved the types I and III response in BALB/c mice and mouse lung epithelial cells (MLE-15) in a neutralization-independent mechanism, likely through binding the CX3C motif [99]. Neutralizing anti-F protein mAb, Palivizumab, did not improve the IFN response. This is expected since the F protein stimulates the production of IFN [99,100]. Studies that examined the human peripheral blood mononuclear cell response to RSV found that pDCs were the primary mediator of IFNα production in a *RIG-1*-dependent manner [101,102]. This response was impaired in infants suggesting that pDCs have a role in responding to RSV infection but are less responsive in young children [103].

It has also been shown that the administration of IFNα to mice before RSV challenge can decrease IL-4Rα and Th2 polarization [104]. In a clinical trial, topical administration of IFNα2a improved symptoms, but had no effect on viral shedding suggesting that viral load is not a correlate of disease severity [105]. Although no side effects were noted in this study, IFNα2a has shown dose-dependent side effects in other contexts including influenza-like symptoms, neurotoxicity [106], and pulmonary toxicity [107]. However, IFNα treatment is generally safe with appropriate caution and monitoring of potential toxicity [105,108,109]. The IFN response of the host undoubtedly influences RSV disease, and infants and children show dysfunctions, indicating the role of the impaired IFN response in severe disease in these groups [64,103,110].

The function of type III IFNs in RSV infection is an area that requires further investigation. Research has shown that in an in vitro model of primary human nasal epithelial cells, RSV infection induced IFNλ1 through *RIG-1* activation [111]. Pretreatment with IFNλ1 resulted in resistance to RSV replication [112]. Surprisingly, no type I IFNs were expressed following RSV infection. However, in primary human airway epithelium cultures, IFNλ1 pretreatment did not inhibit RSV infection [113]. These conflicting results are likely due to differences in the assays used. In a study of young children with RSV it was found that the mRNA levels of IFNλ1-4 were positively correlated with age in the control group of healthy children. However, no correlation was noted in RSV-infected children, possibly due to RSV’s ability to suppress the IFN response [114]. Another study found that type I, II, and III IFN responses increased with age and were lowest in children under 6 months of age [115]. Higher levels of IFNλ2/3 were associated with a reduced risk of hospitalization. However, a separate study of infants with RSV did not find a correlation of IFNλ2/3 with clinical outcomes [116]. Instead, IFNλ1 was associated with an increased clinical severity index and respiratory rate. RSV activation of epidermal growth factor (EGF) receptor causes suppression of IRF1, antagonizing the IFNλ response, mediated by RSV F protein [117]. These studies suggest that type III interferons have a role in the immune response to RSV infection. RSV viral proteins, NS1, NS2, and G protein suppress these antiviral responses.

When infants are infected with RSV, a Th2 polarization is often linked to RSV immunopathology, and the lack of a strong Th1 response in infants and young children can lead to more severe illness [82,118,119]. RSV activates Rab5a GTPase in cells and mice to suppress IRF1-dependent IFNγ. Knockdown of Rab5a increases IFNγ by mediating IRF1 nuclear translocation [120]. Studies have shown that minimal IFNγ production in RSV infected neonatal mice can result in reduced viral clearance and increased disease severity. However, intranasal administration of IFNγ can improve these outcomes by activating alveolar macrophages [121]. Interestingly, blocking IFNγ or depleting NK and T cells has been associated with an increased antibody response in neonatal mice during RSV infection [122], whereas in adult mice, this depletion impairs the antibody response due to CD4 T cell-dependent antibody production [122]. This is likely due to increased viral load in IFNγ-depleted mice, resulting in more antigen exposure. While viral loads were lower in IFNγ-depleted mice, only one viral gene was measured by qRT-PCR and no other factors were measured to determine pathology. These findings suggest that IFNγ is needed for protecting neonatal mice from RSV, and this protection may be independent of T cell-mediated antibody production.

Type I IFNs can promote IFNγ expression by CD8 T cells via STAT4 in conjunction with TCR activation [104] and in NK cells with IL-12 via STAT1 [102,123]. However, IFNß has been shown to suppress DC production of IL-12, which is linked to Th1 differentiation and IFNγ production [124]. These studies suggest that IFNγ is protective during RSV infection and may contribute to increased susceptibility of neonates and children. Overall, these findings support clinical evaluations of infected neonates and children, which have concluded that RSV does not induce a robust IFNγ response [41,42], due to a combination of viral and host factors.

## 6. SARS-CoV-2

COVID-19 is caused SARS-CoV-2. This virus belongs to the family Coronaviradae and has a single-stranded, positive-sense RNA enclosed in an envelope [125]. Its genetic material is made up of 29.9 kb and contains 27 ORFs which encode 31 different proteins, including the four structural proteins (S, E, M, N) [125]. The first two-thirds of the genome encodes two large poly proteins (ORF1a and ORF1ab) which are cleaved by viral proteases to form 16 non-structural proteins (nsp1-16) [126,127,128]. These proteins are involved in replication, transcription, and interfering with the host’s innate defenses. The remaining third of the genome encodes 10 accessory proteins and the four structural proteins mentioned earlier [126,127,128]. As of June 2023, COVID-19 has caused 767 million confirmed cases and 6.9 million deaths worldwide since its emergence in 2019 [129]. 

Similarly to other respiratory viruses, SARS-CoV-2 triggers an IFN response through the recognition of various viral PAMPs by PRRs [130,131]. These include the S protein by TLR4 and TLR2, the E protein by TLR2 [132], and viral ssRNA and dsRNA intermediates by TLR7/8, TLR3, and RLRs [133,134,135,136]. However, similarly to RSV, SARS-CoV-2 has strategies to evade IFN signaling and expression [137,138,139,140]. Several viral proteins including NSP1, ORF6, and NSP13 have been shown to inhibit type I IFNs through different mechanisms including interference with host mRNA translation, nuclear translocation of IRF3 and STAT1, and binding with STAT1 to prevent phosphorylation, respectively [141,142]. ORF6 has also been shown to inhibit type I IFNs by mechanisms that include the inhibition of nuclear translocation of IRF3 and the inhibition of STAT1 nuclear translocation [143,144]. NSP13, which is highly conserved among coronaviruses, has been shown to inhibit type I and II signaling through binding STAT1 and preventing phosphorylation by JAK1 [145].

Several other SARS-CoV-2 viral proteins have been shown to inhibit the IFN response, specifically NSP1, NSP3, NSP5, NSP12, NSP13, NSP14, NSP15, ORF3a, ORF3b, ORF6, ORF7a, ORF7b, ORF8, ORF9b, N, and M reported to inhibit IFNß expression by suppression of RLR-mediated signaling [137,138,142,144]. Further studies indicate ORF7a interferes with TANK-binding kinase (TBK1) preventing IRF3 phosphorylation, and ORF9b interferes with the interaction of MAVS and *RIG-1* [146]. NSP 6 has been shown to interact with TBK1 to inhibit IRF3 activation and STAT1/2 phosphorylation. ORF 7a, ORF 7b, ORF3a, and the M protein were shown to inhibit IFNß in a luciferase reporter assay, a feature attributed to STAT1 and/or STAT2 activation depending on the viral protein [142]. Moreover, using a luciferase reporter, NSP13, NSP14, NSP15 and ORF6 were found to inhibit IFNß by disrupting nuclear translocation of IRF3 [138].

While SARS-CoV-2 primarily replicates in the nasopharyngeal and type II alveolar epithelial cells, recent evidence has shown that ACE2 expression (the receptor for SARS-CoV-2) and TMPRSS2 (a protease required for viral cell entry) are also present in other cells [147]. A mouse model of SARS-CoV-2 was created by expressing hACE2 in transgenic mice originally developed to study SARS-CoV-1 [148]. A mouse-adapted strain has also been developed, which is attributed to a mutation in the receptor binding domain of the S protein [149]. Interestingly, new variants of SARS-CoV-2 (including B.1.1.7) contain the same substitution and can infect mice [150,151]. Different small animal models, such as ferrets and golden hamsters, which are both naturally susceptible to SARS-CoV-2 have varying outcomes from mild to lethal [151].

In contrast to other respiratory viruses, children and neonates are not more likely than adults to develop severe COVID-19, possibly due to differences in IFN signaling [152,153,154,155,156]. Elderly adults with severe COVID-19 have shown lower levels of type I IFNs, potentially due to IFN autoreactive antibodies [157,158]. The other major finding related to age-dependent differences in the IFN response found reduced IFNα secretion by pDCs in adults with severe COVID-19 [159]. The other major finding related to age-dependent differences in the IFN response found reduced IFNα secretion by pDCs in adults with severe COVID-19 [159], and patients with autoimmune polyendocrine syndrome type-1 (APS-1) which produces autoantibodies against type I IFNs are at increased risk of severe disease [158], and require mechanical ventilation [160]. pDCs and fibroblasts of these patients indicated blunted IFN responses [161]. Age-related discrepancies in the IFN response to SARS-CoV-2 can be attributed to several plausible mechanisms related to suppression of the IFNs. Several studies have shown a role of IFNs in immune-mediated pathology with a deficient early response followed by a persistent and heightened late response associated with severe disease. ScRNASeq analysis of bronchial alveolar lavage (BAL) or lung tissue samples identified a type I IFN-associated inflammatory signature, consistent with findings in peripheral blood, suggesting that an early and regulated IFN response is protective while a latent dysregulated response is pathogenic [162,163].

The findings suggest that the neonatal immune system changes rapidly in early years of life affecting the IFN response. Neonates are less likely to suffer severe outcomes from SARS-CoV-2 infection, but pre-term infants are at higher risk of COVID-19 and pre-term birth due to medically induced birth.

## 7. Influenza Virus

Influenza virus is an enveloped ssRNA virus belonging to the Orthomyxoviridae family [164]. It has a segmented genome with eight segments encoding various proteins including polymerase subunits, surface glycoproteins, nucleoprotein, matrix protein, membrane protein, nuclear export protein, and nonstructural proteins [164,165,166]. An analysis from 1996 to 2016 estimated a suggested 32.1 million influenza virus episodes and 5.7 million hospitalizations in adults per year globally with those over 65 years of age having the highest rate of hospitalization [167]. In 2018, an analysis of children under 5 years of age estimated 10.1 million cases of influenza-like respiratory illness with 870,000 hospital admissions [168]. Pandemic strains have a much greater impact than seasonal strains with the 1918 pandemic resulting in estimates of 40–50 million deaths globally [169]. Age is an important factor contributing to disease severity, with the young and old particularly susceptible to severe disease. One of many factors contributing to severity by strains of influenza is their ability to suppress the IFN response [170].

Influenza virus activates TLRs 3, 7, and 8, as well as *RIG-1*/MAVs, inducing the production of type I and III IFNs [171,172]. The influenza NS1 protein suppresses the IFN response through multiple mechanisms [173,174] including blocking host protein translation by interfering with pre-mRNA processing via binding to cleavage and polyadenylation specificity factor 30 (CPSF30), a component of the pre-mRNA processing machinery responsible for 3′ cleavage and polyadenylation [175,176]. NS1 also prevents dsRNA-mediated IFN responses by scavenging dsRNA and blocking IRF3 phosphorylation [177]. Additionally, NS1 inhibits *RIG-1*-mediated IFN production by preventing TRIM25 multimerization and CARD domain ubiquitination [178]. Influenza PB1 and PA can also counteract the type I IFN response in mice with PA interacting with IRF3 to prevent its activation and PB1-F2 interfering with the MAVs adapter protein [170,179,180]. PB2 interacts with MAVs and IPS-1 to reduce IFNß transcription, and NS1 is the dominant IFN antagonist of influenza virus, with IFN suppression being associated with more severe disease [181,182,183]. IFNß is protective [72,184], but type I IFNs can contribute to mortality [185]. IFNα and IFNλ reduce viral load but only IFNλ controls viral replication in the upper airways [186].

Neonatal mice are more susceptible to influenza virus due to poor IFN responses [187]. Reduced IFNγ response in young mice delays viral clearance and increases mortality [188]. Adoptive transfer of adult CD8 T cells into neonatal mice confers resistance to influenza challenge dependent on IFNγ [189]. A separate study utilizing IFNγ knockout mice found no difference in mortality of WT and IFNγ-/- mice after IAV challenge, while finding increased antigen-specific T cells in IFNγ-/- mice [190]. In mice, type III IFNs were found to have an important role in the course of infection with IFNλ levels found to be higher in the lungs of influenza-infected BALB/c mice than type I IFNs [191]. Children infected with influenza virus had higher levels of IFN in their nasal wash than children infected with RSV, but that IFN was not associated with reduced viral shedding. Children between the ages of 29 and 54 months who were given a live attenuated influenza vaccine (LAIV) had upregulated *ISGs* and lower viral loads on days 2 and 7, which was attributed to asymptomatic respiratory virus infection prior to the administration of LAIV [192]. Children with an autosomal recessive IRF7/IRF9 deficiency are at greatest risk for severe influenza due to poor IFN responses [66,193,194]. In one report, three children with a rare TLR3 deficiency were found to be susceptible to severe influenza pneumonitis [193]. It was shown that treatment with IFNα2a and IFNλ1 rescued this susceptibility using pulmonary epithelial cells differentiated from patient-derived induced pluripotent stem cells (iPSCs) [193]. Taken together, these studies corroborate the importance of IFN responses to influenza, particularly in infants, children, and the elderly.

## 8. Conclusions

IFNs have an early and crucial role in the body’s antiviral response. Three major respiratory viruses affecting humans, i.e., RSV, influenza virus, and SARS-CoV-2, have IFN antagonistic features (summarized in Figure 1) [1,102,195,196]. For RSV, the *NS1*, *NS2*, and *G* genes are key effectors in this response [102], while for SARS-CoV-2 the *NSP1*, *ORF6*, and *NSP13* genes have a particularly potent effect on IFN. Influenza virus has the *NS1* gene as the canonical IFN antagonist [137], but antagonism activities are associated with the *PA*, *PB1*, *PB1-F2*, and *PB2* genes [170].

In infants and young children, type I, II, and III IFN responses are lower compared to adults [41,42]. The reduced IFNα producing pDCs and reduced expression of IL-12, (a cytokine that increases production of IFNγ) are principally linked to the diminished IFNγ [44]. Additionally, a general lack of Th1-type responses causes IFNγ to be diminished due to several mechanisms such as a lack of IL-12 production in neonatal moDCs [56], IL-4 induced apoptosis of neonatal Th1 cells and increased expression of GATA3 [58], and increased IL-4 expression [60]. While IFNs are protective against RSV [105,115,121] and influenza [192,193] in neonatal mice and children, they experience more severe disease compared to older counterparts due to blunted IFN responses. However, SARS-CoV-2-infected infants and young children are not at increased risk for severe disease compared to healthy adults [152,153,154,155]. There is evidence that dysregulation of IFNs may contribute to severe disease, but that an early regulated response is protective [197,198].

A beneficial IFN response is associated with lower viral loads, faster viral clearance, and reduced disease severity [66]. IFN treatments have been investigated for viral infections, including approved use of IFNα2a for hepatitis C [199]. However, excessive inflammation has been linked to higher levels of IFNs [198] highlighting the importance of properly regulated anti-viral responses and caution with off-target effects and side effects [199]. Type I and type III IFNs have distinct roles, with type III IFN demonstrating a more targeted role and generally mediating a more diminished inflammatory response [200]. Drugs that boost endogenous IFN have also been investigated as treatments for viral infection [201,202,203]. The lack of a strong interferon response in neonates and children has been associated with susceptibility to influenza and RSV, but this has not been observed in the context of SARS-CoV-2 infection [155]. The protective and pathogenic role of IFNs in respiratory virus infection is likely linked to underlying mechanisms.

## Figures and Tables

**Figure 1 microorganisms-11-02179-f001:**
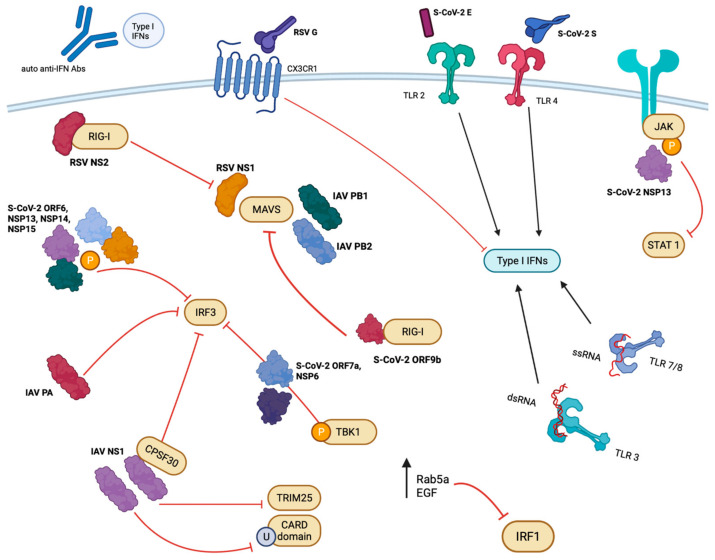
Overview of viral protein interactions. Viral proteins from RSV, SARS-CoV-2, and IAV antagonize or agonize IFN responses using a variety of mechanisms. An abridged illustration of the viral proteins and host factors involved is pictured summarizing those mentioned throughout the text. RIG-I (retinoic acid-inducible gene 1), EGF (epidermal growth factor), TLR (toll-like receptor), JAK (Janus kinase), MAVS (mitochondrial antiviral-signalizing protein), TRIM25 (tripartite motif-containing protein 25), CARD (caspase recruitment domains), CPSF30 (cleavage and polyadenylation specificity factor subunit 4), IAV (influenza A virus), RSV (respiratory syncytial virus), S-CoV-2 (SARS-CoV-2), NS/NSP (non-structural/non-structural protein), IRF (interferon regulatory transcription factor), TBK1 (TANK-binding kinase), U (ubiquitination), P (phosphorylation). Created with BioRender and Microsoft PowerPoint.

## Data Availability

Data sharing not applicable.

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
