# Peer review of "Interferons—Implications in the Immune Response to Respiratory Viruses"

_microorganisms, 2023, doi:10.3390/microorganisms11092179_

Round 1

Reviewer 1 Report

The authors describe IFNs, IFN-signaling pathways and their roles in three major respiratory viral infections, with a focus on young children. The apparent differences between adults and young children and neonates are important to keep in mind when studying the role of IFNs in studies on pathogenesis and/or treatment.

My major concern is that the information is too condensed. Almost every paragraph introduces a number of proteins, genes or signaling events. It's hard to see the forest for the trees. The back-and-forth between the different types of IFNs is not easy to follow.

I think that the manuscript would greatly benefit from a few figures. For example, separate figures on the function/signaling pathways for the three types of IFNs, possibly also indicating how the three respiratory viruses interfere with the pathways. I would also suggest to explore whether the readability would be better if the information on each type of IFN would be kept together, probably up to the section on 'Mice as a model'.

There a few sentences that should be checked (see below), but with a substantial rewrite of the manuscript those sentences may have been changed already.

Check sentence ‘To reduce IFN effector function and result in unregulated viral replication and disease pathogenesis viruses have developed various strategies to antagonize the JAK/STAT pathway [25].’ on page 2.

The sentence ‘When stimulated, cord blood responses showed decreased IFNα expression in whole blood and monocytes from adult donors.’ on page 3 is confusing. Is the reduced expression in the cord blood samples or in the ‘whole blood and monocytes from adult donors’?

Suggestion to change the order of the two paragraphs in 'Mice as a model', with first a description of IFN expression and then the part on Mx1.

On page 6: 'TL4' is 'TLR4'

Reviewer 2 Report

This is a well-written review that does not contain any figures or tables.

Specific comments,

The section starting with “The primary function of ISGS…. In response to viral DNA and RNA, the IFIT family of ISG…”- Why only mention the IFIT family when there are so many additional ISGs with the capacity to limit viral RNA and DNA replication? Suggest to add additional ISGs in this section.

The authors could consider adding a Table or (Figure) to summarize some of their key points for example a Table with references regarding young children and their interferon responses to these respiratory viruses pointing towards protection or a more severe outcome, depending on the study.

Only found one misspelling- TL4 should be TLR4 second paragraph SARS-CoV-2 section.

Section Influenza virus “TLRs” spelled out again. Please check so that abbreviations explained first time used-perhaps the order of some sections were changed.

Round 2

Reviewer 1 Report

Thanks you for addressing my comments